# Residential and wealth-related disparities of high fertility preferences in Ethiopia: A decomposition analysis

Melaku Birhanu Alemu[1,2]*, Ayal Debie[1,3], Samrawit Birhanu Alemu[4], Gizachew A. Tessema[2,5]

1 Department of Health Systems and Policy, Institute of Public Health, University of Gondar, Gondar, Ethiopia, 2 Curtin School of Population Health, Curtin University, Perth, Australia, 3 College of Medicine and Public Health, Flinders University, Adelaide, Australia, 4 Department of Public Health, Debre Markos University, Debre Markos, Ethiopia, 5 enAble Institute, Curtin University, Bentley, Perth, Western Australia, Australia

* melakubt@gmail.com

**Data Availability Statement:** The Ethiopian demographic health survey (DHS) dataset is a third party data which is collected and owened by Measure DHS website http://www.measuredhs.

## Abstract

### Background

Fertility preference significantly influences contraceptive uptake and impacts population growth, especially in low and middle-income countries. In the previous pieces of literature, variations in fertility preference across residence and wealth categories and its contributors were not assessed in Ethiopia. Therefore, we decomposed high fertility preferences among reproductive-aged women by residence and wealth status in Ethiopia.

### Methods

We extracted individual women's record (IR) data from the publicly available 2016 Ethiopian Demographic Health Survey (EDHS) dataset. A total of 13799 women were included in the study. Multivariate decomposition analysis was conducted to identify the factors that contributed to the differences in the percentage of fertility preferences between rural and urban dwellers. Furthermore, we used an Erreygers normalized concentration index and curve to identify the concentration of high fertility preferences across wealth categories. The concentration index was further decomposed to identify the contributing factors for the wealth-related disparities in high fertility preference. Finally, the elasticity of wealth-related disparity for a change in the socioeconomic variable was estimated.

### Results

The weighted percentage of women with high fertility preference among rural and urban residents was 42.7% and 19%, respectively, reflecting a 23.7 percentage point difference. The variations in fertility preference due to the differences in respondents' characteristics accounted for 40.9%. Being unmarried (8.4%), secondary (14.1%) and higher education (21.9%), having more than four children (18.4%), having media exposure (6.9%), middle (0.4%), richer (0.2%) and richest (0.1%) wealth were the positive and city administration

com. The data can be requested from DHS. Anyone can get access to the data by mentioning the purpose of data use. However, the data cannot be shared without consent from DHS.

**Funding:** The author(s) received no specific funding for this work.

**Competing interests:** The authors have declared that no competing interests exist.

(-30.2%), primary education (-1.3%) were the negative contributing factors for the variations in high fertility preferences due to population composition. Likewise, about 59% of the variations in fertility preference were due to variations in coefficients. City administration (22.4%), primary (7.8%) and secondary (7.4%) education, poorer wealth (0.86%) were the positive and having media exposure (-6.32%) and being unmarried (-5.89%), having more than four children (-2.1%) were the negative factors contributing to the difference in high fertility preferences due to the change in coefficients across residents. On the other hand, there was a pro-poor distribution for high fertility preferences across wealth categories with Erreygers normalized concentration index of ECI = -0.14, SE = 0.012. Having media exposure (17.5%), primary (7.3%), secondary (5.4%), higher (2.4%) education, being unmarried (8%), having more than four children (7.4%), rural residence (3%) and emerging (2.2%) were the positive and city administration (-0.55) was the negative significant contributor to the pro-poor disparity in high fertility preference.

## Conclusion

The variations in high fertility preferences between rural and urban women were mainly attributed to changes in women's behavior. In addition, substantial variations in fertility preference across women's residences were explained by the change in women's population composition. In addition, a pro-poor distribution of high fertility preference was observed among respondents. As such, the pro-poor high fertility preference was elastic for a percent change in socioeconomic variables. The pro-poor high fertility preference was elastic (changeable) for a percent change in each socioeconomic variables. Therefore, women's empowerment through education and access to media will be important in limiting women's desire for more children in Ethiopia. Therefore, policymakers should focus on improving the contributing factors for the residential and wealth-related disparities in high fertility preferences.

## Introduction

Africa has a fast growing population in the world [1]. The high rate of population growth could contribute to low health outcomes by limiting basic life needs [2]. A rapid population increase could potentially decrease per capita income if the economy fails to keep pace with the population growth rate [3]. High fertility has a negative effect on worsening unemployment rate, increasing family size, and risks of maternal and child health outcomes [4]. The risk of maternal mortality is higher during the first pregnancies and fifth and subsequent births [5] which could contribute to the 5.4 million annual deaths of children less than 5 years [6]. Despite the Sustainable Development Goal (SDG) focusing on lessening under-five mortality rates to below 25 deaths per 1000 live births by the end of 2030 [7], under-five mortalities are still alarmingly high in sub-Saharan Africa (SSA) and South Asia [6]. To curve the current population growth, understanding fertility choice is important to realize the demographic transitions of various nations [8–10].

Nearly 90% of the differences between countries in actual fertility are accounted for solely by differences in desired fertility [11]. Fertility preferences reflect women's or couples' desire for children which could have a direct impact on contraception demand [10, 12] and

population growth [11]. In 2015, more than 90% of developing countries with rapid population growth instituted policies to check the growth rate [3]. Fertility rates in most low- and middle-income countries (LMICs) are declining at a slower pace [13–15]. The Ethiopian population is still characterized by a high fertility rate [16]. Over the years, the fertility rates for Ethiopia were 4.21 births per woman in 2019, 4.12 births per woman in 2020, and 4.01 births per woman in 2021 [17].

Fertility desire can be affected by multiple factors, such as place of residence, wealth status, the number of alive children, and women education [18–22]. The perceived cost and value of children could affect fertility preferences and they thought that children in LMICs were perceived to have high economic importance for the households [23]. On the contrary, women empowerment, urban residence, and better education achievements were associated with low fertility preferences [18, 19]. Substantial variation across countries in the mean ideal number of children and as of the first survey, the means ranged from a high of 8.3 for Chad to a low of three for Lesotho, and overall, the unweighted average was nearly six children per woman in SSA [24]. Similarly, the average ideal number of children in Ethiopia was also about five per woman in 2016 [24].

Various policies and strategies have been implemented globally to promote sexual and reproductive health and reduce fertility rates. These include raising the age of marriage or union formation and the age of first birth, enhancing access to modern contraceptives, and increasing the time interval between births [25]. The Ethiopian government also adopted the population policy. It developed different strategies to decrease fertility, including raising the minimum age of marriage to 18, empowering women through education, and increasing contraceptive distribution services [26, 27]. However, the population policy has faced challenges like budgetary constraints, a lack of monitoring and evaluation, and a lack of a comprehensive population program [27].

Variations of fertility preference across women's residence and household wealth status were reported in the previous literature [18–22]. In Ethiopia, 51.2% of rural and 64.5% of poorest household women had a fertility desire of five or more children per woman; however, only 25.8% of urban and 25.7% of richest household women had a fertility desire of five or more children per woman in 2016 [28]. Despite significant variations that have been observed across women's residence and wealth status, the potential contributing factors for the differences in percentage change of high fertility preference across residence and wealth status have not been well identified. Therefore, this study aimed to assess Ethiopia's residential and wealth-related disparities in high fertility preference.

The study holds practical significance for policymaking, healthcare planning, and socioeconomic interventions. The findings can guide targeted population policies and family planning programs, optimizing resource allocation based on disparities across regions and wealth categories. The study could inform the importance of women's empowerment in shaping fertility preferences and guiding the design of population health interventions. Overall, the study provides a crucial foundation for informed decision-making and interventions aimed at fostering sustainable demographic transitions in Ethiopia.

## Methods and materials

### Ethiopia's healthcare delivery system

Ethiopia's health service is structured into a three-tier system: primary, secondary, and tertiary levels of care. A primary health care unit (PHCU) comprises four health centers (HCs), five health posts within each health center, and a primary hospital. Each health post is responsible for a population of 3,000–5,000 people. A health center provides both preventive and curative

services. In addition to what an HC can provide, a primary hospital provides emergency surgical services, including cesarean section, and gives access to blood transfusion services. This primary health care offers SRH services, including ANC, PNC, delivery, family planning and adolescent and youth-friendly services [29, 30]. The secondary level of care consists of general hospitals. In addition, it serves as a referral center for primary hospitals. Finally, the tertiary level of care comprises federally-run, specialized hospitals and university hospitals [31, 32]. The secondary and tertiary level hospitals provide RH services and referral cases in both inpatient and ambulatory [29]. The mode of delivery is according to the life cycle of women by ensuring the continuum of care starting from preconception to postpartum and neonatal period and through women's reproductive health [29].

## Study settings and data source and study population

Ethiopia is located in the horn of Africa and is the 2nd populous country in Africa with an estimated population of over 117 million in 2021 [33]. This study was conducted using the EDHS 2016 data set collected in the nine administrative regions, namely Tigray, Afar, Amhara, Benishangul-Gumuz, Gambela, Harari, Oromia, Somali, and Southern Nations, Nationalities, and Peoples of Region (SNNP), and two city administrative regions (Addis Ababa and Dire-Dawa,). More than 80% of the country's total population lives in the regional states of Amhara, Oromia, and SNNP [34]. A total of 13799 respondents were used for the analysis. However, to investigate the role of maternal health services-related variables, we limited our analysis to samples with this information. Therefore, since maternal health services information were collected for women with a birth in the last 5 years preceding the survey, a total of 6134 women were included in the study population.

## Measurements of variables

In this study, considering the average total fertility rate in Ethiopia (4.46 children/woman), we defined high fertility preference as a woman's desire to have more than or equal to five children. However, when a woman desired less than five children, she was regarded as having a low fertility preference [35]. The ideal number of children a woman prefers to have in her lifetime (v613) was used as a fertility preference measure.

The region variable was categorized into developed (Tigray, Amhara, Oromia, and SNNP), emerging (Afar, Benishangul Gumuz, Gambela, and Somali) and city administration (Addis Ababa, Dire-Dawa, and Harari). Even though Harari is not a city administration in Ethiopian regional classification, the composition of the population (urban-rural) is more similar to city administrations than emerging regions. Therefore, we included the region as a city administration [36]. Media exposure was assessed when a woman read a magazine/newspaper or listen to the radio or watch television [37].

The socioeconomic level of respondents was measured using a wealth index. The wealth index was constructed by using a principal component analysis for the urban and rural residents separately. Wealth scores are assigned to households based on the quantity and kind of consumer goods they own, which can range from a television to a bicycle or automobile, as well as home attributes including flooring materials, restroom amenities, and sources of drinking water [38]. It was categorized into five quintiles (poorest, poorer, middle, richer, and richest). The elasticity of wealth-related disparity and percentage contribution of independent variables for the observed socioeconomic variation for high fertility preference were computed. Wealth-related disparity is considered elastic if a 1% change in an independent variable could increase/decrease the wealth-related disparity by more than 1%.

## Data analysis

The data were weighted using sampling weight, primary sampling unit, and strata before any statistical analysis to restore the representativeness of the survey. We cleaned and recoded the extracted data for analysis. Respondents who prefer to have more than 16 children were excluded from this study as having more than 16 children in a lifetime is not common. All the analyses were conducted by applying sampling weights.

Multivariate decomposition analysis was used to model factors for the observed fertility preference differences according to geographic residence. The disparity in fertility preference between the urban and rural residences in Ethiopia could be attributed to the composition of the population (endowment) and the change in the characteristics (coefficients) of the explanatory variables between the residents. Therefore, fertility preference disparities are additively decomposed to the endowments and coefficients of characteristics. The non-linear decomposition analysis model utilizes the output of logistic regression. Even though the Oaxaca-Blinder approach is the well-known multivariate decomposition method, especially for linear outcomes, the application of this method for a non-linear outcome (rate and proportions) has pitfalls. Therefore, a multivariate decomposition for nonlinear models was applied. The model can partition the differences in high fertility preference in urban and rural residents into sociodemographic and economic characteristics which can affect the disparities [39].

The logistic model of preference disparities between urban and rural residents is given as follows: [40]

$$Logit\ (A) - Logit\ (B) = F(X_A\beta_A) - F(X_B\beta_B) = \underbrace{[F(X_A\beta_A) - F(X_B\beta_A)]}_{E} + \underbrace{[F(X_B\beta_A) - F(X_B\beta_B)]}_{C}$$

The "E" component refers to the part of the differential owing to differences in endowments or population composition (characteristics). The "C" component refers to that part of the differential attributable to differences in coefficients or effects. The analysis was done by using, a *mvdcmp* STATA command for the non-linear decomposition analysis [40].

We used Erreygers normalized concentration index (ECI) and concentration curve to present wealth-related disparities in high fertility preference [41–44]. However, the methods could not tell us the partition of the different socioeconomic factors to the observed difference [45]. Therefore, a separate decomposition of Erreygers normalized concentration index was performed to assess the elasticity and percent contribution of independent variables for the propoor disparity in high fertility preference.

A concentration curve is the relative measure of wealth-related disparities and it plots the cumulative percentage of high fertility preference against the cumulative percentage of the population ranked by socioeconomic status. The concentration curve above the diagonal (line of equality) shows the high fertility preference concentration among low socioeconomic status women (pro-poor distribution). Similarly, the concentration curve below the line of equity shows the high fertility concentration among high socioeconomic status (pro-rich distribution) [45].

The concentration index is twice the area between the concentration curve and the 45-degree line of equality. If there is no wealth-related disparity for high fertility preference, the value would be 0. However, if there is a pro-poor socioeconomic disparity the value would be negative [45]. For the unbound variables, the concentration index value lies between -1 and 1. The concentration curve is mathematically defined as:

$$C = \frac{2}{\mu}COV(h, r)$$

Where: C is the concentration index; μ is the mean health variable (proportion of high fertility preference); COV: covariance (h is the high fertility preference; r is the rank of an individual to wealth index) [45].

However, for the bound variables (high/low fertility preference), the concentration index ranges from μ −1 to 1− μ [46]. Erreygers suggested a modification in the concentration curve to account for the bound nature of the variable [44, 47–49].

$$ECI = 4*\mu*CI(y)$$

Where: ECI: Erreygers normalized concentration index; μ: mean health variable (proportion of high fertility preference); CI(y): generalized concentration index.

The Erreygers normalized concentration (ECI) was further decomposed to identify factors contributing to wealth-related disparities in high fertility preference. The elasticity of disparity for a change in independent variables was estimated. The elasticity is the change observed in the disparities when there is a one percent change in the variables. The analysis was done by using, "svy" STATA command [50].

## Ethics approval and consent to participate

We requested the Ethiopian demographic health survey (DHS) dataset from the Measure DHS (http://www.measuredhs.com). All the data underlying the findings described in the manuscript is included in the manuscript. A written informed consent was obtained from all participants for inclusion in the study.

## Results

### Sociodemographic and economic characteristics of respondents

A total of 13962 weighted reproductive-aged women were included in the final study. Among the participants, 76.6% (N = 10,689) of women were from rural areas. Around 40% of the respondents were aged 15–24 years, with comparable distributions for women from urban and rural areas. On average, respondents from the urban area were more educated. For instance, 30% and 20% of women in the urban areas completed secondary and higher education. However, only 7.1% and 1.3% of women completed secondary and higher education, respectively. Similarly, the urban residents are wealthier than respondents from rural areas. For example, 93.1% of women in urban areas are from the richest quantile however only 8.1% of rural residents are in the richest quintile (Table 1).

### Fertility preferences across residence and wealth status

On average, urban residents prefer to have 3.81 (95%CI: 3.74–3.89) children, and rural residents prefer to have 4.58 (95% CI: 4.53–4.64) children. Nearly 20% of urban and 42.7% of rural residents have a high fertility rate. Similarly, the poorest quantile preferred to have 5.3 (95% CI:5.2–5.4) children, while the poorest quantile only prefers 3.9 (95%CI:3.8–3.9) children. The majority (53.4%) of the poorest respondents have a high fertility preference. Only 21.9% of the richest women have high fertility preferences (Fig 1).

### Fertility and fertility preferences

The majority of urban (70.6%) and rural (57.5%) respondents wanted another child anytime in the future. Nearly 20% of respondents in the urban area and approximately one-third (32%) from the rural area did not want to have a child in the future. While only 5.6% of mothers in the urban area have more than five children, over a quarter (25.9%) of respondents from the

**Table 1. Socioeconomic and demographic characteristics of respondents, 2016 (n = 13962).**

| Variables | Category | Urban (n = 3273, 23.44%) | Rural (n = 10689, 76.56%) | Percentage point differences (Δ%) |
|---|---|---|---|---|
| **Age (years)** | 15–24 | 43.55 | 40.57 | 2.98 |
| | 25–35 | 39.06 | 36.59 | 2.47 |
| | 36–49 | 17.39 | 22.84 | -5.45 |
| **Region** | Developed | 65.46 | 95.32 | -29.86 |
| | Emerging | 4.33 | 4.33 | 0.00 |
| | City admin | 30.21 | 0.35 | 29.86 |
| **Religion** | Orthodox | 61.05 | 38.35 | 22.7 |
| | Muslim | 17.81 | 33.55 | -15.74 |
| | Protestant | 20.20 | 25.78 | -5.58 |
| | Other | 0.95 | 2.32 | -1.37 |
| **Current marital status** | Married | 44.71 | 67.58 | -22.87 |
| | Unmarried | 55.29 | 32.42 | 22.87 |
| **Women educational status** | No education | 14.86 | 54.12 | -39.26 |
| | Primary | 33.10 | 37.44 | -4.34 |
| | Secondary | 30.04 | 7.13 | 22.91 |
| | Higher | 22.00 | 1.31 | 20.69 |
| **Women occupation** | Not working | 38.01 | 53.92 | -15.91 |
| | Professional | 8.16 | 0.90 | 7.26 |
| | Clerical | 4.13 | 0.07 | 4.06 |
| | Sales | 27.74 | 11.34 | 16.4 |
| | Agriculture | 3.10 | 25.11 | -22.01 |
| | Services | 6.87 | 2.40 | 4.47 |
| | Skilled manual | 6.20 | 3.03 | 3.17 |
| | Unskilled manual | 2.03 | 1.15 | 0.88 |
| | Other | 3.77 | 2.09 | 1.68 |
| **Wealth index** | Poorest | 2.28 | 19.44 | -17.16 |
| | Poorer | 0.76 | 22.76 | -22 |
| | Middle | 1.05 | 24.64 | -23.59 |
| | Richer | 2.84 | 25.10 | -22.26 |
| | Richest | 93.06 | 8.05 | 85.01 |

rural area had more than five children. Ideally, a quarter of respondents from the urban area and half of the respondents from rural areas would prefer to have more than 5 children (Table 2).

### Maternal health services

The maternal health service-related variables applied to 6134 women for their recent children. Among those, institutional delivery showed the highest difference among residents with a 60.26 percentage point's gap. Similarly, there is a 35.83 percentage point difference in complete antenatal care attendance across residents (Table 3).

### Fertility preference across the residence

There is a statistically significant difference in high fertility preference between urban and rural residents in one or more groups of the explanatory variables. Table 4 summarizes the residential distribution of high fertility preference.

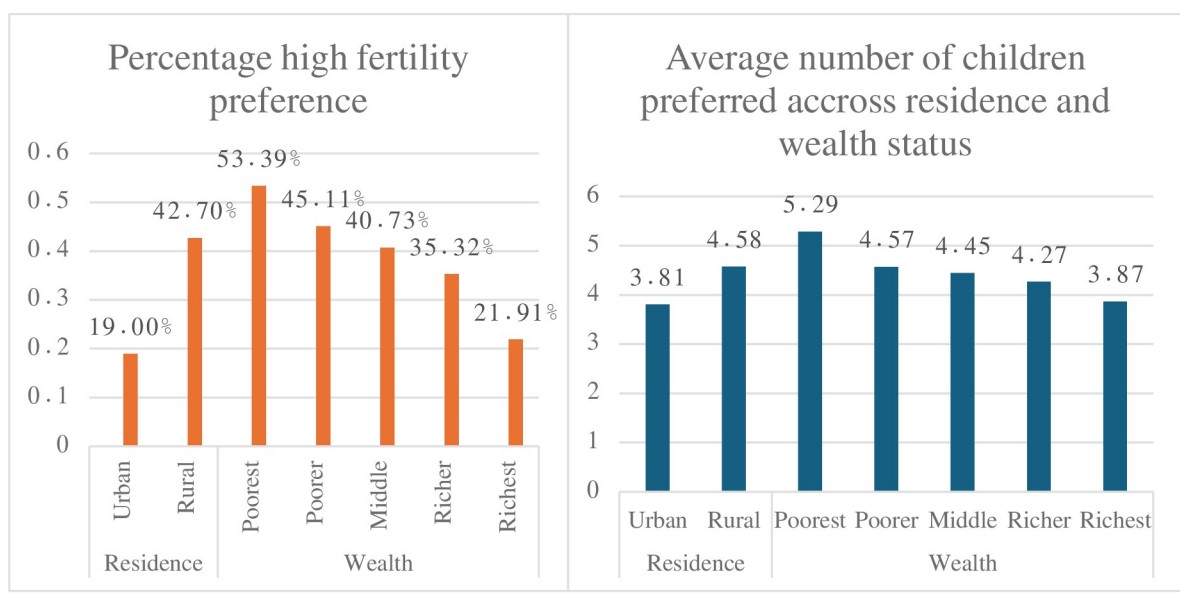

**Fig 1. Percentage of high fertility and average number of children preferred across residence and wealth status.**

## Decomposition of fertility preference across the residence

More than half (59.1%) of the disparity in high fertility preference was attributed to the difference in the effect (coefficients) of each variable for urban and rural residences and the remaining disparity (40.9%) in high fertility preference between rural and urban residents was explained by the changes in population composition (endowments). There was a significant difference in the composition of women's age, marital status, region, education, parity media exposure and wealth. Region, education and parity showed a significant difference due to effects (Table 5).

Equalizing the number of secondary and higher educated rural women to the urban level could decrease the observed rural-urban child preference gap by 14%, and 22%, respectively. If rural residents had a similar behavioral response to primary education as urban residents, the rural-urban child preference gap would expect to drop by 7.8%. This means that the effect of primary education on reducing the high child preference was not as strong for urban residents

**Table 2. Percentage distribution of fertility and fertility preferences of the urban and rural respondents.**

| Variables | Category | Urban (n = 3273, 23.44%) | Rural (n = 10689, 76.56%) | Percentage point differences (Δ%) |
|---|---|---|---|---|
| **Parity** | ≤ 4 Children | 92.68 | 68.95 | 23.73 |
| | >4 Children | 7.32 | 31.05 | -23.73 |
| **Number of alive children** | ≤ 4 Children | 94.30 | 74.20 | 20.1 |
| | >4 Children | 5.70 | 25.80 | -20.1 |
| **Desire for another child** | Wanted | 70.61 | 57.52 | 13.09 |
| | Not wanted | 21.35 | 32.05 | -10.7 |
| | Undecided | 6.63 | 9.02 | -2.39 |
| | Infertile/Sterilized | 1.41 | 1.42 | -0.01 |
| **Ideal number of children** | ≤ 4 Children | 81.05 | 57.33 | 23.72 |
| | > 4 Children | 18.95 | 42.67 | -23.72 |

**Table 3. Maternal health services characteristics of respondents who for their recent child, 2016 (n = 6576).**

| Variables | Category | Urban (n = 879, 13.37%) | Rural (n = 5697, 86.63%) | Percentage point differences (Δ%) |
|---|---|---|---|---|
| ANC visit | 0 | 9.33 | 40.64 | -31.31 |
| | 1–3 | 26.78 | 31.30 | -4.52 |
| | ≥4 | 63.88 | 28.05 | 35.83 |
| Mode of delivery | Vaginal delivery | 87.67 | 98.86 | -11.19 |
| | Cesarean section | 12.33 | 1.14 | 11.19 |
| Place of delivery | Home delivery | 14.43 | 73.09 | -58.66 |
| | Health facility | 85.34 | 25.08 | 60.26 |
| | Other | 0.23 | 1.84 | -1.61 |

as they are for rural residents. Residing in a city administration explained -30.15% of the endowment's differences for the high desired fertility between the rural and urban. Similarly, the city administration is the second largest predictor explaining the preference gap of endowments following the wealth characteristics. Matching the wealth status of the middle, richer and richest rural residences to the urban level would lower the gap in high child preference by 0.35%, 0.2%, and 0.11%.

## The cumulative contribution to the residential disparity

Education (34.7%) explained the majority of rural-urban disparities in fertility preference due to the composition of characteristics. Similarly, region, parity, marital status, and media exposure contributed to -30.2%, 18.4%, 8.4%, and 6.9% of the residential disparities in fertility preference as a result of the endowment. The majority (22.2%) of the disparities in fertility

**Table 4. Percentage change (Δ%) of high fertility preference across residence among women in Ethiopia, EDHS 2016 (n = 5802).**

| Variables | Category | Urban (n = 1287, 22.18%) | Rural (n = 4515, 77.82%) | Percentage point differences (Δ%) | P-value (Chi-square) |
|---|---|---|---|---|---|
| Age (years) | 15–24 | 36.52 | 29.72 | 6.80 | <0.0001 |
| | 25–35 | 37.92 | 40.80 | -2.88 | |
| | 36–49 | 25.56 | 29.48 | -3.92 | |
| Current marital status | Married | 56.72 | 78.98 | -22.26 | <0.0001 |
| | Unmarried | 43.28 | 21.02 | 22.26 | |
| Region | Developed | 16.94 | 49.97 | -33.03 | <0.0001 |
| | Emerging | 32.01 | 41.37 | -9.36 | |
| | City admin | 51.05 | 8.66 | 42.39 | |
| Education | No education | 31.47 | 69.86 | -38.39 | <0.0001 |
| | Primary | 35.12 | 25.51 | 9.61 | |
| | Secondary | 20.12 | 4.03 | 16.09 | |
| | Higher | 13.29 | 0.60 | 12.69 | |
| Parity | ≤ 4 Children | 79.41 | 55.02 | 24.39 | <0.0001 |
| | >4 Children | 20.59 | 44.98 | -24.39 | |
| Media exposure | No | 25.49 | 76.19 | -50.70 | <0.0001 |
| | Yes | 74.51 | 23.81 | 50.70 | |
| Wealth index | Poorest | 4.35 | 42.83 | -38.48 | <0.0001 |
| | Poorer | 1.79 | 19.56 | -17.77 | |
| | Middle | 2.25 | 17.25 | -15.00 | |
| | Richer | 4.74 | 14.80 | -10.06 | |
| | Richest | 86.87 | 5.56 | 81.31 | |

**Table 5. Factors contributing to fertility preference across residences in Ethiopia, EDHS 2016.**

| Variable | Category | Endowments (E) | | Coefficients (C) | |
|---|---|---|---|---|---|
| | | β (Std. Err.) | Percent (%) | β (Std. Err.) | Percent (%) |
| Age in years | 15–24 | Ref | | | |
| | 25–35 | -0.15 (0.03) ** | -0.64 | 0.62 (0.94) | 2.61 |
| | 36–49 | 0.61 (0.09) *** | 2.57 | 0.12 (0.51) | 0.51 |
| | Subtotal | | 1.93 | | 3.12 |
| Current marital status | Married | Ref | | | |
| | Unmarried | 1.99 (0.27) *** | 8.4 | -1.40 (1.17) | -5.89 |
| Region | Developed | Ref | | | |
| | Emerging | 0.00084 (0.00008) *** | 0 | -0.04 (0.17) | -0.18 |
| | City admin | -7.2 (2.15) *** | -30.15 | 5.31 (2.32) ** | 22.39 |
| | Subtotal | | -30.15 | | 22.21 |
| Education | No education | Ref | | | |
| | Primary | -0.3 (0.05) *** | -1.26 | 1.85 (0.84) ** | 7.81 |
| | Secondary | 3.4 (0.53) *** | 14.07 | 1.75 (0.99) ** | 7.39 |
| | Higher | 5.2 (0.97) *** | 21.91 | -1.12 (1.21) | -4.73 |
| | Subtotal | | 34.72 | | 10.49 |
| Parity | ≤ 4 Children | Ref | | | |
| | >4 Children | 4.37 (0.35) *** | 18.42 | -0.50 (0.24) ** | -2.12 |
| Media exposure | No | Ref | | | |
| | Yes | 1.64 (0.52) *** | 6.93 | -1.50 (2.25) | -6.32 |
| Wealth | Poorest | Ref | | | |
| | Poorer | 0.0011(0.008) | 0.00 | 0.20 (0.54) ** | 0.86 |
| | Middle | 0.084 (0.018) *** | 0.35 | 0.22 (0.69) | 0.93 |
| | Richer | 0.047 (0.013) *** | 0.20 | 1.16 (0.74) | 4.88 |
| | Richest | 0.025 (0.053) ** | 0.11 | -0.45 (0.8) | -1.91 |
| | Subtotal | | 0.66 | | 4.76 |
| Constant | | | | 9.02 (5.49) | 32.84 |

***, **, * = = > Significant at 1%, 5%, 10% level; Std. Err., Standard Error; Ref, Reference category; E, Difference due to endowment; C, Difference due to coefficient.

preference were explained by the difference in the coefficient of the region followed by education (10.5%) (Fig 2).

## Concentration curve and index

The wealth-related disparities in the ideal number of children showed a significant pro-poor distribution. This means that the poorest women prefer to have five or more children than the wealthiest women. The finding was supported by the negative Erreygers normalized concentration index (ECI = -0.14, 95%CI: -0.16, -0.12) (Fig 3).

## Decomposition of fertility preference across wealth status

Various socioeconomic factors explained the pro-poor distribution of high fertility preference among reproductive-aged women. Media exposure and education are the dominant factors explaining Ethiopia's pro-poor distribution of fertility preference.

Having media exposure (17.5%), primary (7.3%), secondary (5.4%), higher (2.4%) education, being unmarried (8%), having more than four children (7.4%), rural residence (3%) and emerging (2.2%) were the positive and city administration (-0.55) was the negative significant

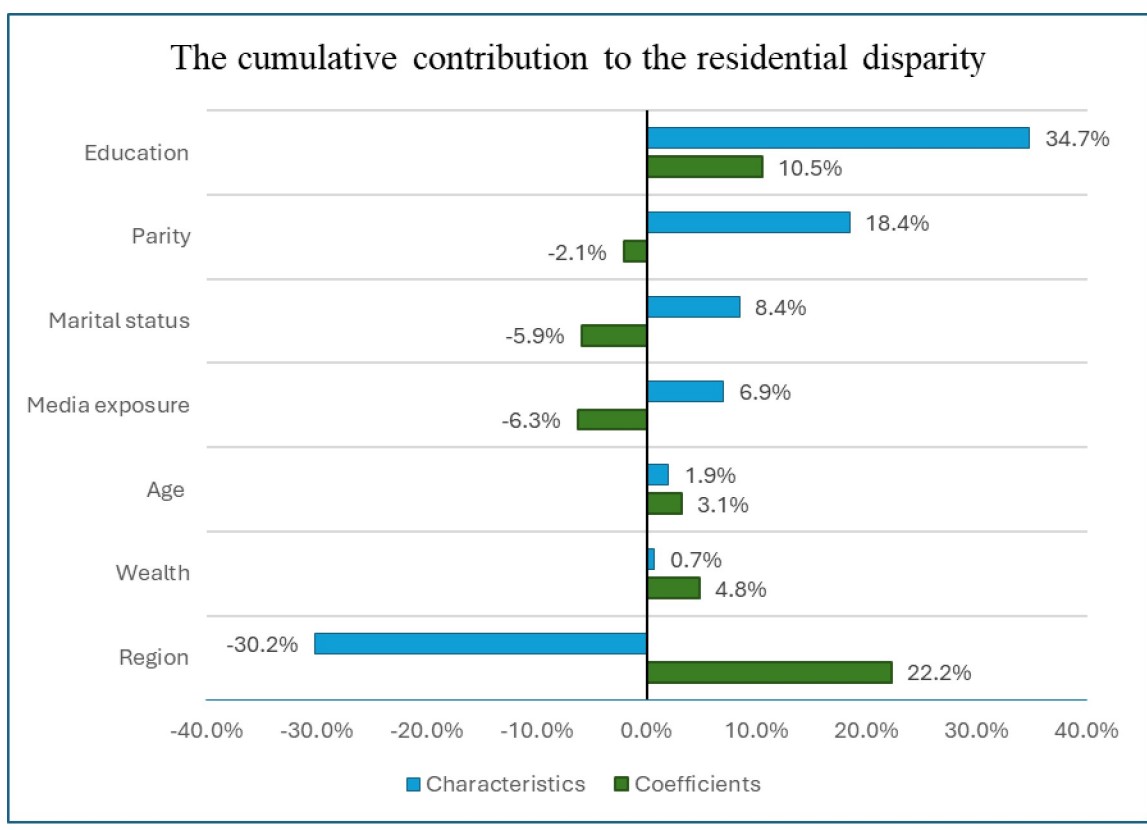

**Fig 2. Percentage contribution of socioeconomic variables explained by the difference in the composition (characteristics) and coefficients of variables for rural-urban high fertility preference disparity among reproductive-aged women in Ethiopia, EDHS 2016.**

contributor to the pro-poor disparity in high fertility preference among reproductive-aged group women in Ethiopia. Therefore, increasing media exposure among people with low economic status could lower the pro-poor distribution of high fertility by 17.5. Likewise, increasing the primary education of women with low financial situations could reduce the observed high fertility preference disparity by 7.32% (Table 6).

The wealth-related disparity for fertility preference is elastic for a percentage change in sociodemographic factors. The wealth-related disparity was highly elastic for a change in parity (0.19). A 1% change in parity (from 1–4 children to more than five children) could increase the wealth-related disparity in fertility preference by 19%. Similarly, A 1% increase in education from no education to primary, secondary, and higher education could decrease wealth-related disparity by 12%, 8.3%, and 4.3% respectively. Regarding media exposure, a 1% change in media exposure status could decrease the pro-poor disparity by 7.9% (Table 6).

## Discussion

This study examined the residential and wealth-related disparity in fertility preference among reproductive-aged women and identify the contributing factors for the observed rural/urban fertility preference gap and pro-poor high fertility distribution. The residential decomposition analysis showed 40.93% of the variation in high fertility preference between urban and rural residents is attributed to the composition of characteristics. A variety of socio-demographic, fertility-related, and wealth-related variables showed a significant composition variation which

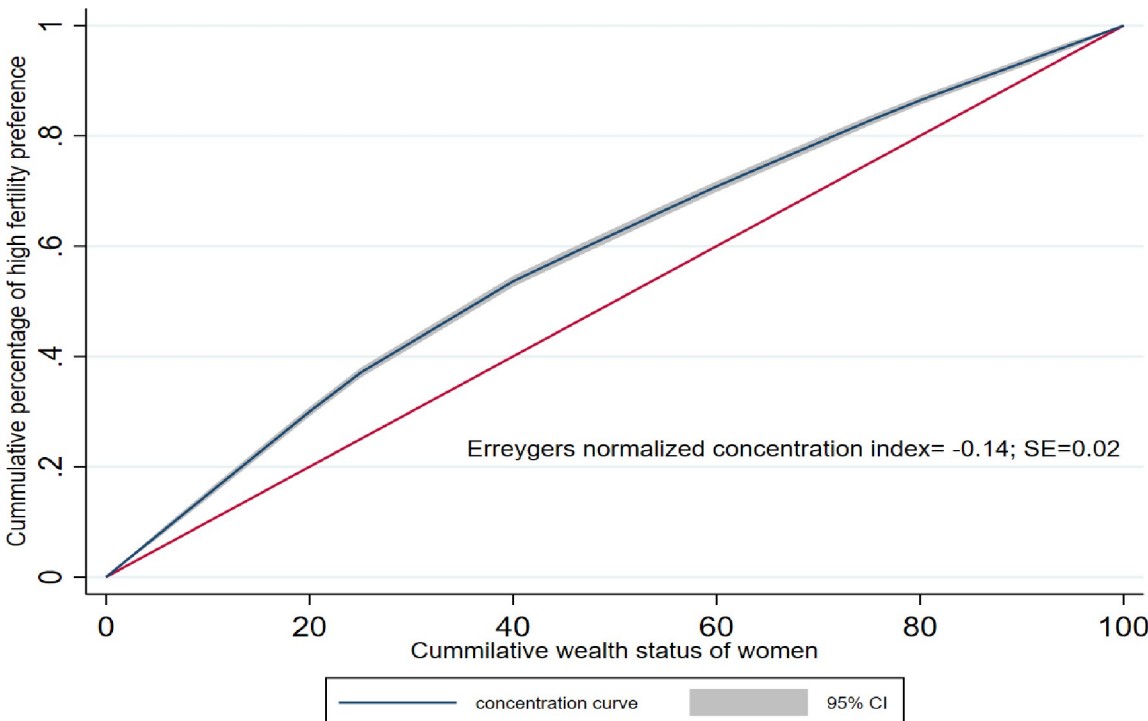

**Fig 3. Erreygers normalized concentration curve of wealth-related disparities for high fertility preferences among women aged 15–49 in Ethiopia, EDHS 2016.**

could affect the high fertility preference gap between rural and urban residents. A significant pro-poor distribution of high child preference was observed among women in the study population.

In our study, education was the highest contributor to the rural-urban fertility preference disparity as a result of the composition of characteristics. About 22% of the disparity in fertility preference between rural and urban residents was attributed to the composition of women who completed higher education. This means that if we educate the rural women to the level of urban women, the observed preference gap could reduce by 22%. A similar study conducted in Ethiopia reviled the association of maternal education with the preferences for a fewer number of children [51]. Similarly, a study conducted in Pakistan reviled that illiteracy was significantly associated with the idealization of a large number of children [52]. Likewise, a variety of studies from Ethiopia [51, 53], Bangladesh [54], Egypt [55], and India [56] showed an important association of education with fertility. Therefore, policymakers should focus on improving the education level of women to lower the preferences for having a high number of children and consequently a high fertility rate.

We also observed that 30.15% of the difference in women's high fertility preference gap in urban and rural were due to differences in the composition of city administration residents. The high percentage change in city administration could be explained by the increased rate of urbanization. A study conducted in India indicated that culture and demographic characteristics could shape fertility behavior [56]. Similarly, a study conducted in Bangladesh found that urbanization has a significant role in fertility reduction [54]. On top of population composition, nearly 22.4% of the variation was explained by the effect of coefficient in urban and rural residences. This could be explained by the behavioral difference in fertility [56].

**Table 6. Wealth-related disparities factors for high child fertility preference in Ethiopia.**

| Variable | Category | dF/dx (Std. Err.) | Elasticity | Concentration Index | Absolute contribution | Percentage contribution |
|---|---|---|---|---|---|---|
| **Age in years** | 15–24 | Ref | | | | |
| | 25–35 | -0.0030 (0.012) | 0.065 | -0.033 | -0.0021 | 1.50 |
| | 36–49 | 0.012 (0.015) | 0.076 | 0.0066 | 0.00050 | -0.35 |
| | Subtotal | | | | -0.0016 | 1.15 |
| **Residence** | Urban | Ref | | | | |
| | Rural | 0.085(0.013)*** | 0.26 | -0.016 | -0.0043 | 3.03 |
| **Current marital status** | Married | Ref | | | | |
| | Unmarried | -0.098 (0.01) *** | -0.11 | 0.10 | -0.011 | 7.95 |
| **Region** | Developed | Ref | | | | |
| | Emerging | 0.2(0.012) *** | 0.049 | -0.081 | -0.004 | 2.77 |
| | City admin | 0.1 (0.014) *** | 0.007 | 0.11 | 0.00078 | -0.55 |
| | Subtotal | | | | -0.0048 | 2.22 |
| **Education** | No education | Ref | | | | |
| | Primary | -0.13 (0.011) *** | -0.12 | 0.085 | -0.010 | 7.32 |
| | Secondary | -0.2 (0.014) *** | -0.083 | 0.093 | -0.0077 | 5.38 |
| | Higher | -0.22 (0.016) *** | -0.043 | 0.079 | -0.0034 | 2.36 |
| | Subtotal | | | | -0.021 | 15.06 |
| **Parity** | ≤ 4 Children | Ref | | | | |
| | >5 Children | 0.27 (0.013) *** | 0.19 | -0.056 | -0.011 | 7.40 |
| **Media exposure** | No | Ref | | | | |
| | Yes | -0.078 (0.012) *** | -0.079 | 0.32 | -0.025 | 17.46 |

***, **, * = = > Significant at 1%, 5%, 10% level; dF/dx, discrete change of dummy variable from 0 to 1; Std. Err., Standard Error; Ref, Reference category.

Parity (18.42%) and marital status (8.4%) women explained differences in fertility preference in rural and urban residents due to the composition of characteristics. On average, multipara women are stated to have high fertility. Their child preference could contribute to having multiple children or there is social desirability bias in answering the ideal number of children. For example, if a mother has 6 children at the time of the interview, answering the ideal number of children less than 6 could indirectly mean one or more children is unplanned. Parity and fertility preference showed a clear association in this dataset [57]. Similarly, shreds of evidence showed that marital status including the age of marriage significantly affects the parity of a woman [58, 59] and it can shape the reproductive and sexual behavior of women and men [59–61].

Media exposure composition explained around 7% of variations in rural and urban fertility preference variation. Increasing access to media of rural residents at the urban level could lower the observed rural and urban fertility gap by 7%. A study conducted in Bangladesh showed that media exposure has a significant association with low fertility [62]. In addition, it contributed to 17.5% of the pro-poor concentration of high fertility preference. A 1% increase in media exposure (magazine, newspaper, radio, or television) could lower wealth-related disparity by 7.9%. Media exposure is found to explain different health-related wealth-related disparities in different pieces of literature [47, 48].

Equalization of the distribution of wealth of rural households to the middle, richer and richest urban households would lower the observed gap by 0.35%, 0.2%, and 0.11%, respectively. Generally, fertility preference showed a pro-poor distribution. This means that high fertility preference was concentrated in low-income households. The reason could be the cultural and educational difference between the poor and the rich households. Furthermore, the cost of the

child of a poor community is usually shared with the community and the children could help the economy of the household which is in line with the backer quality-quantity model [63].

Media exposure (17.5%), education (15.1%), marital status (8%), and parity (7.4%) were the major factors that explained the pro-poor distribution of high fertility preferences. The wealth-related disparity of high fertility preference was elastic for a change in one of the explanatory factors. For instance, a 1% change in education from no education to primary education could lower the pro-poor disparity by 13%. A variety of literature showed a similar characteristic for the elasticity of wealth-related disparity for a change in an education [47, 48].

### Limitation of the study

We have used the "ideal number of children" as a measure of fertility preference. However, we acknowledge that an ideal number of children can be affected by the current number of children that women already have. In addition, the DHS used a durable property to compute the wealth index of the households. However, household's consumption could give a good estimation of the wealth index. Furthermore, we only accounted for the fertility preference of women, but fertility preference could be affected by the preference of partner.

### Conclusion

The variations in high fertility preferences between rural and urban women were mainly attributed to changes in women's behavior. In addition, substantial variations in fertility preference across women's residences were explained by the change in women's population composition. In addition, a pro-poor distribution of high fertility preference was observed among respondents. As such, the pro-poor high fertility preference was elastic for a percent change in socioeconomic variables. The pro-poor high fertility preference was elastic (changeable) for a percent change in each socioeconomic variables.

Policymakers need to address the contributing factors to the high fertility preference in rural areas and lower socioeconomic settings to prevent disproportionate population growth. To achieve this, education programs should be designed to enhance educational attainment among women. It is also important to develop region-specific strategies that take into consideration the diverse socio-demographic characteristics. Furthermore, media campaigns should be organized, particularly targeted at women with lower education levels and wealth status to effectively lower high fertility preference in Ethiopia. The finding could be applicable to similar context with a high fertility rate.

### Acknowledgments

We would like to acknowledge Professor David Lindsay for his input in the writing of the article.

### Author Contributions

**Conceptualization:** Melaku Birhanu Alemu, Ayal Debie, Gizachew A. Tessema.

**Data curation:** Melaku Birhanu Alemu.

**Formal analysis:** Melaku Birhanu Alemu, Ayal Debie, Samrawit Birhanu Alemu, Gizachew A. Tessema.

**Methodology:** Melaku Birhanu Alemu, Ayal Debie, Samrawit Birhanu Alemu.

**Software:** Samrawit Birhanu Alemu.

**Visualization:** Melaku Birhanu Alemu, Samrawit Birhanu Alemu, Gizachew A. Tessema.

**Writing – original draft:** Melaku Birhanu Alemu, Samrawit Birhanu Alemu.

**Writing – review & editing:** Melaku Birhanu Alemu, Ayal Debie, Gizachew A. Tessema.

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
