## [Decision Letter · Decision Letter 0]

4 Dec 2023

PONE-D-23-22238Residential and wealth-related disparities for high fertility preferences and it’s contributing factors among reproductive-aged women in Ethiopia: a decomposition analysisPLOS ONE

Dear Dr. Alemu,

Thank you for submitting your manuscript to PLOS ONE. After careful consideration, we feel that it has merit but does not fully meet PLOS ONE’s publication criteria as it currently stands. Therefore, we invite you to submit a revised version of the manuscript that addresses the points raised during the review process.

Please submit your revised manuscript by Jan 18 2024 11:59PM. If you will need more time than this to complete your revisions, please reply to this message or contact the journal office at plosone@plos.org. Please include the following items when submitting your revised manuscript:A rebuttal letter that responds to each point raised by the academic editor and reviewer(s). You should upload this letter as a separate file labeled 'Response to Reviewers'.A marked-up copy of your manuscript that highlights changes made to the original version. You should upload this as a separate file labeled 'Revised Manuscript with Track Changes'.An unmarked version of your revised paper without tracked changes. You should upload this as a separate file labeled 'Manuscript'.If applicable, we recommend that you deposit your laboratory protocols in protocols.io to enhance the reproducibility of your results. Protocols.io assigns your protocol its own identifier (DOI) so that it can be cited independently in the future. For instructions see: https://journals.plos.org/plosone/s/submission-guidelines#loc-laboratory-protocols. Additionally, PLOS ONE offers an option for publishing peer-reviewed Lab Protocol articles, which describe protocols hosted on protocols.io. Read more information on sharing protocols at https://plos.org/protocols?utm_medium=editorial-email&utm_source=authorletters&utm_campaign=protocols.

We look forward to receiving your revised manuscript.

Kind regards,

Abay Woday Tadesse, 

Academic Editor

PLOS ONE

Journal Requirements:

Additional Editor Comments:

PONE-D-23-22238

"Residential and wealth-related disparities for high fertility preferences and it’s contributing factors among reproductive-aged women in Ethiopia: a decomposition analysis"

Thank you for submitting your interesting paper for publication in the PLOS ONE journal. Generally, the manuscript is well-designed; however, I have a few concerns that need to be addressed before formal acceptance for publication.

Conclusion and recommendations:

The provided recommendations are broad and might pose challenges for relevant authorities to implement based on the study's findings. Therefore, it would be beneficial if the authors could provide specific recommendations directed towards the relevant stakeholders.

The implication of the findings:

Summarize the practical significance of your findings. For instance, you can summarise how the observed findings impact existing practices or policies. This would consider the relevance and potential consequences of this study's findings.

Further, you could discuss the generalizability of your findings to other populations or settings. This could clarify the extent to which your results can be applied beyond the specific context of your study.

Reviewers' comments:

Reviewer's Responses to Questions

**Comments to the Author**

1. Is the manuscript technically sound, and do the data support the conclusions?

Reviewer #1: Yes

Reviewer #2: Yes

2. Has the statistical analysis been performed appropriately and rigorously? 

Reviewer #1: Yes

Reviewer #2: Yes

3. Have the authors made all data underlying the findings in their manuscript fully available?

Reviewer #1: Yes

Reviewer #2: Yes

4. Is the manuscript presented in an intelligible fashion and written in standard English?

Reviewer #1: Yes

Reviewer #2: Yes

5. Review Comments to the Author

Reviewer #1: The manuscript is generally interesting and with good insight. The methdology section specfically data analysis technique was interesting and able to answer the research quastions It will be better enough for comments and simply track sections if it could have page and line numbers.

Reviewer #2: Thank you for the opportunity to review this highly relevant manuscript. I enjoyed reading it. The manuscript is well designed.

Here below, I’ve highlighted my comments to improve the paper prior to publication.

Overall, it would be great if the authors put the page numbers and line numbers to easily cite the areas of concern.

A. Abstract

1. The recommendations are general and could be difficult to take action by concerned bodies based on the findings of this study. Hence, it would be good if the authors put specific recommendations to relevant stakeholders (policymakers, planners, programmers and practitioners) to alleviate such issue.

B. Main document

Introduction

1. The authors put a good background to the current study. It would be also great to put the existing initiatives by the government to address the problem in a paragraph (including policy, strategy, plans and service access, etc). For example, National population policy and reproductive health strategy of the country could be consulted.

Measurements of variables in the methods section

1. The categorization of the region variable needs reference citation. It could have effect on both residence and wealth status indicators.

2. The wealth index concept and its construction need more elaboration (particularly what it intends to measure and what type of data was used in its calculation). You could visit the DHS Program regarding wealth index construction and also cite the country’s 2016 DHS survey methodology.

Data analysis

1. Why the authors preferred to use the Erreygers normalized concentration index (ECI) in the presence of other similar indices such as the Wagstaff’s concentration index?

2. What STATA command did the authors use to calculate the concentration index? It would be worthwhile mentioning it to the reader.

Results

Decomposition of fertility preference across the residence

1. “More than half (40.9%) of variations in high fertility preference between rural and urban residents were explained by the difference in composition (endowments) and the remaining differences (59.1%) were attributed to the difference in the effect of each variable for urban and rural residences (coefficients)”. The term “more than half” should be revised as 40.9% is not comparable with that.

Concentration curve and index

1. It’s good reporting the values of concentration index with its standard error. It would be also great to incorporate the 95% confidence interval of the concentration index.

Conclusions

1. See the comment above in the abstract section related to the recommendations.

6. PLOS authors have the option to publish the peer review history of their article (what does this mean?). If published, this will include your full peer review and any attached files.

Reviewer #1: No

Reviewer #2: No

---

## [Author Response · Author response to Decision Letter 0]

24 Jan 2024

Dear Editor and reviewers,

Thank you for your comments and suggestions. We have revised the manuscript and attached the track change and point by point response. Please let us know if you have further comments.

Best,

Melaku

---

## [Decision Letter · Decision Letter 1]

12 Feb 2024

Residential and wealth-related disparities of high fertility preferences in Ethiopia: a decomposition analysis

PONE-D-23-22238R1

Dear Dr. Alemu%,

We’re pleased to inform you that your manuscript has been judged scientifically suitable for publication and will be formally accepted for publication once it meets all outstanding technical requirements.

Kind regards,

Abay W. Tadesse

Academic Editor

PLOS ONE

Additional Editor Comments (optional):

Reviewers' comments:

Reviewer's Responses to Questions

**Comments to the Author**

1. If the authors have adequately addressed your comments raised in a previous round of review and you feel that this manuscript is now acceptable for publication, you may indicate that here to bypass the “Comments to the Author” section, enter your conflict of interest statement in the “Confidential to Editor” section, and submit your "Accept" recommendation.

Reviewer #1: All comments have been addressed

Reviewer #2: All comments have been addressed

2. Is the manuscript technically sound, and do the data support the conclusions?

Reviewer #1: Partly

Reviewer #2: Yes

3. Has the statistical analysis been performed appropriately and rigorously? 

Reviewer #1: Yes

Reviewer #2: Yes

4. Have the authors made all data underlying the findings in their manuscript fully available?

Reviewer #1: Yes

Reviewer #2: Yes

5. Is the manuscript presented in an intelligible fashion and written in standard English?

Reviewer #1: Yes

Reviewer #2: Yes

6. Review Comments to the Author

Reviewer #1: The revision is well done and the revised document is looking good. It is better to focuse on residential and wealth-related inequalities in all the sections of the manuscript.

Reviewer #2: Thank you for the opportunity to review this paper. My previous comments have been well addressed and I recommend for the publication of this manuscript.

7. PLOS authors have the option to publish the peer review history of their article (what does this mean?). If published, this will include your full peer review and any attached files.

Reviewer #1: No

Reviewer #2: **Yes: **Yihalem Abebe Belay

---

## [Editor Report · Acceptance letter]

27 Feb 2024

PONE-D-23-22238R1 

PLOS ONE

Dear Dr. Alemu, 

I'm pleased to inform you that your manuscript has been deemed suitable for publication in PLOS ONE. Congratulations! Your manuscript is now being handed over to our production team.

Kind regards, 

on behalf of

Mr. Abay Woday Tadesse 

Academic Editor

PLOS ONE